# Once Resin Composites and Dental Sealants Release Bisphenol-A, How Might This Affect Our Clinical Management?—A Systematic Review

**DOI:** 10.3390/ijerph16091627

**Published:** 2019-05-09

**Authors:** Anabela Baptista Paula, Debbie Toste, Alfredo Marinho, Inês Amaro, Carlos-Miguel Marto, Ana Coelho, Manuel Marques-Ferreira, Eunice Carrilho

**Affiliations:** 1Institute of Integrated Clinical Practice, Institute for Clinical and Biomedical Research (iCBR), area of Environment Genetics and Oncobiology (CIMAGO), CNC.IBILI, Faculty of Medicine, University of Coimbra, 3000-075 Coimbra, Portugal; mig-marto@hotmail.com (C.-M.M.); anasofiacoelho@gmail.com (A.C.); eunicecarrilho@gmail.com (E.C.); 2Institute of Integrated Clinical Practice, Faculty of Medicine, University of Coimbra, 3000-075 Coimbra, Portugal; debbie_toste@hotmail.com (D.T.); alfredomarinho1996@hotmail.com (A.M.); ines.amaros@hotmail.com (I.A.); 3Institute of Endodontics, Institute for Clinical and Biomedical Research (iCBR), area of Environment Genetics and Oncobiology (CIMAGO), CNC.IBILI, Faculty of Medicine, University of Coimbra, 3000-075 Coimbra, Portugal; m.mferreira@netcabo.pt

**Keywords:** bisphenol A, dental sealants, endocrine disruptor, environmental levels, exposure, monomers, prevention, resin composites

## Abstract

(1) Background: Bisphenol A (BPA) based dental resins are commonly used in preventive and reparative dentistry. Since some monomers may remain unpolymerized in the application of dental resin, they dissolve in the saliva. (2) Methods: The literature search was carried out in Pubmed, Cochrane and Embase databases. Randomized controlled trials, cohort studies and case-control studies that evaluated BPA levels in human urine, saliva and/or blood were included. (3) Results: The initial search had 5111 results. A total of 20 studies were included in the systematic review. Most studies showed an increase of the levels of bisphenol A 1 h after treatments with composite resins and dental sealants. One week after treatments the levels were decreased. (4) Conclusions: Some clinical precautions should be taken to decrease the release of BPA, namely the use of rubber dam, the immediate polishing of all resins used, or the use of glycerin gel to avoid non-polymerization of the last resin layer, and mouthwash after treatment. Another preventive measure in addition to the above-mentioned is the use of the smallest possible number of restorations or sealants, a maximum of four per appointment. These measures are even more important in children, adolescents and pregnant women.

## 1. Introduction

Endocrine active substances (EAS) such as Bisphenol A (BPA) (2,2-bis[4′-(2′hydroxy-3′-methacryloxy) phenyl] propane) and BPA-derivatives can cause estrogenic activity, and may affect human health [1,2].

This synthetic chemical compound was first synthesized by an acid-catalyzed reaction of phenol and acetone in 1891 [3,4]. Today, BPA is used in the manufacture of many types of products, including polycarbonate plastic and epoxy resin presented in bottles, food packaging, toys, cars, detergents, pesticides and dental resins materials [5,6].

Food are cited as the primary source of BPA exposure in humans [7,8]. However, some recent studies with dental materials and with other nonfood sources suggested that are many other origins of BPA that may contribute to cumulative exposure in humans [4,9].

In dentistry, monomers with a BPA-core are commonly used in resin-based materials such as root canal sealers, adhesives, composites and sealants [2].

Although dental materials typically do not contain pure BPA, this compound can be the result of the manufacturing process or a byproduct of degradation of bisphenol A-glycidyl methacrylate (bis-GMA) or other components such as Ethoxylated bisphenol A dimethacrylate (BisEMA), bis-dimethylaminopropyl (BisDMA), 2,2-bis-(4-(3-methacryloxypropoxy)phenyl)propane (BisPMA), and bisphenol a diglycidyl ether (BADGE) [9,10].

In the intraoral environment, these materials are exposed to extreme thermal changes, pH variances, mechanical erosion, and degradation occurrence from bacterial and salivary enzymes, which can cause BPA release [5,6]. During or just after resin placement, its leaching can also occur by incomplete monomer polymerization [5,6].

There are some studies in the literature that have demonstrated the presence of BPA in human saliva, urine and blood after application of resin dental materials. The possibility of this chemical substance being absorbed systemically through the blood should be a concern to oral health care professionals [5,6,11].

BPA was recognized by an endocrine disruptor that mimics estrogen and alters hormonal function as early as the 1930s [4]. The increase emphasis on BPA release can be attributed to the fact that it plays a role in the pathogenesis of several endocrine disorders, including female and male infertility, hormone dependent tumors such as breast and prostate cancer, polycystic ovary syndrome precocious puberty, several metabolic disorders including obesity, and teratogenic effects, even at a low dose [11].

Even more unbound BPA may be available in vivo due to the fact that BPA has low affinity to protein binding, increasing its estrogenic potential regarding laboratory studies [1].

From all leached and detected ingredients from resin-based materials, BPA has led to the most controversy due to its endocrine disrupting nature [2]. Thus, the objective of this study was to conduct a systematic review of the literature in order to evaluate the release of BPA and its derivatives from the different dental materials and to examine the potential risks to health associated with BPA exposure, answering the following question:

PICO (problem, intervention, comparison and outcome) question: Can the release of BPA after the use of composite resins and/or dental sealants, increase much higher than the acceptable daily exposure, causing harmful effects on the health of children, adolescents and pregnant adults?

## 2. Materials and Methods

### 2.1. Search Strategy

To perform this systematic review and meta-analysis, PRISMA guidelines were used [12,13]. In the present study, three electronic databases were used: Pubmed, Cochrane and Embase. The research included English, Spanish and Portuguese Language filters, using a combination of the keywords such as dental materials, composite resins, dentures, fissure sealants, BPA-derivatives, bisphenol A, and bisphenol A-glycidyl methacrylate. Additional search methods included the reference lists of relevant studies scrutinized manually. Three independently reviewers scrutinized the studies based on the inclusion criteria. A fourth reviewer was consulted where there was uncertainty regarding eligibility, and a decision arrived at by consensus.

### 2.2. Inclusion and Exclusion Criteria

Thus, the inclusion criteria for selection and extraction of data were: (1) randomized controlled trials, cohort studies and case-control studies; (2) tests for evaluation of BPA levels in urine, saliva and/or blood; (3) tests on humans; and (4) tests on adults and children from 3 years of age. Exclusion criteria were (1) in vitro studies; (2) in vivo studies with animals; (3) in vivo assays measuring BPA levels in skin patches. Editorials, reports of cases, letters, comments, personal communications, procedures and studies with insufficient information were excluded.

### 2.3. Data Extraction

The studies that fulfilled the inclusion criteria were processed for the extraction of data. The data were as follows: the name of the first author, the year of publication, the type of study, the sample number, the age of the patients, type of treatment, object of study, BPA sources/materials, BPA evaluation methods, follow-up, results and notes. The extraction of the information was done by three independent authors using a standard form. A consensus meeting was always held to confirm the agreement and to resolve disagreement among the reviewers.

### 2.4. Quality Assessment

The evaluation of the methodological quality of the included studies is essential for understanding the results of the systematic review. This quality of each randomized controlled trial (RCT) study was assessed using the bias risk assessment tool described in the Cochrane Handbook of Systematic Reviews of Interventions (Version 5.1.0) [14]. Briefly, six domains were evaluated: (1) random sequence generation to select the participants (selection bias); (2) allocation concealment (selection bias); (3) blinding intervention of participants and personnel (performance bias); (4) blinding of outcome assessment (detection bias); (5) incomplete outcome data (attrition bias); (6) selective reporting (reporting bias); and (7) other biases. The quality of the case-control study and the retrospective cohort study was evaluated according to the Methodological Index for Non-Randomized Studies (ROBINS-I) [15]. For this index, seven domains were evaluated in three phases: (1) at pre-intervention, biases due to confounding and biases in selection of participants into the study were scrutinized; (2) at intervention, only biases in classification of interventions were scrutinized; (3) at post-intervention, bias due to deviations from intended interventions, biases due to missing data, biases in measurement of outcomes and biases in selection of the reported results were scrutinized.

## 3. Results

This systematic review was registered in PROSPERO with the ID122957.

### 3.1. Study Characteristics

The flow diagram of study selection is shown in Figure 1 [13,16]. A total of 5110 studies were identified through the search in the referred databases, with one added from other sources and, after the removal of duplicates, there were 4232 studies. After sorting by title, 278 studies were obtained. There was a total of 29 studies articles sorted for eligibility after reading the abstract, with 249 being excluded. Reading the full text led to the exclusion of nine records when submitted to the scrutiny of the exclusion criteria. After reading the articles in full, 20 articles were included for the systematic review.

The 20 clinical studies included in the systematic review are described in detail in Table 1. The studies included were mostly RCTs, about 16, one was a case-control study and three were retrospective cohort studies. The sample sizes are completely different, ranging from 4 to 1001 patients, with a mean of 171.60 ± 268.19 (SD). The age of patients ranges from childhood to adults of 55 years old. Several sources of BPA were used such as adhesives, resin composites, dental sealants and acrylic resins to make several types of treatments like restorations, fissure sealants or bonded orthodontic appliance. For evaluating the release of BPA after these treatments, the investigators used three study objects: 15 studies analysis the BPA levels in saliva, four in blood, and eight in urine. The BPA evaluation method of choice was high-pressure liquid chromatography in most studies, but gas chromatography, Enzyme-Linked Immunosorbent Assay (ELISA), estrogenic assay, and flow cytometry (immune and renal function) were also used. The follow-up periods were similar, with evaluations immediately after the treatment, in the first hour and first day. Later follow-up ranged from 1 month to 5 years.

All studies of salivary content showed an increase in the levels of bisphenol A in the evaluations realized immediately after the treatments, either with composite resins or with sealants, within 1 h [1,5,17,18,20,23,26,28,29]. This increase in BPA in most studies ranges from 2 to 42 ng/mL [1,5,18,23,26], although there are some that reports values of 120.05 to 931 ng/mL [17,29]. In other follow-ups, the levels decrease over time, like pre-treatment after 1 week [1,5,17,18,20,23,26,28,29]. Some studies have evaluated the levels of bisphenol A according to the number of surfaces restored or sealed, with an exponential increase from six surfaces [6,9,19]. On the other hand, a study performed the evaluation after the treatment followed by mouthwash, demonstrating an abrupt decrease in levels. [20] Studies describing the types of monomers of the materials used show that some of them have higher levels than others, namely bis-DEMA and BPAHPE [14,15]. Some studies using the Denton^®^ fissure sealant (Dentsply Sirona, York, PA, USA) also have much higher levels of Bisphenol A in saliva [1,19,27].

Only two studies evaluated levels of bisphenol A in the blood and it was not detected in serum at any of the study times [10,19]. However, two studies using blood samples evaluated immune and renal function [30,31]. The immune function is changed at 6 months and 1 year, but not at 5 years with changes in B cell activation, and in monocyte and neutrophil function. These alterations were not associated with resin composites [30]. Renal function was not altered at any time of study [31].

In studies evaluating urinary content, the levels of bisphenol A immediately after treatment increase slightly but not as markedly as with saliva [1,5,17,18,22]. However, there is also a higher increase when the number of surfaces is greater than six. In this study, a slight increase was observed when three to dive surfaces were performed, but a rubber dam was used in 51.6% of the patients [22]. Another study reports BPA increases of about 20% to 25% in children who had between seven and 16 sealants [9].

In the only study where the estrogenic assay was performed, an increase immediately after treatment from 0.1 to 1.43 ppm was observed, with only one type of fissure sealant (Delton^®^) decreasing to levels below 0.1 ppm after 24 h [27]. The study of Raghavan et al. in 2017 [25] analyzes the release of BPA after removal of the orthodontic appliance and the placement of a polymerized acrylic resin retainer in three different ways. The vacuum retainers release more BPA, followed by the chemically processed ones. Heat polymerization shows the lowest release of BPA [25].

### 3.2. Methodological Quality Assessment of Included Studies

The results of the quality assessment of RCTs of the systematic reviews can be seen in Table 2. Nine studies presented flaws in the methodological description of random sequence generation with insufficient information. The allocation concealment was not done in one study and in 10 others it is not properly explained. The performance bias, with blinding of participants and investigators in clinical procedures was impossible in most studies, especially with investigators, since the different physical characteristics of materials make them easy to distinguish from each other. Blinding evaluation of the results was possible with minimal risk of bias in most studies; however, some studies did not have enough information on the intervention bias. Only one study had incomplete outcome data, only with graphical information, without quantitative values. Reporting bias and other biases were minimal risks in all scrutinized studies.

Four studies were not RCTs: one was a case-control study, and three were retrospective cohort studies. The evaluation of quality assessment through the risk of bias was done with the tool ROBINS-I (“Risk of bias in non-randomized studies—Of interventions”), as recommended by Cochrane. The results of this evaluation can be seen in Table 3. The pre-intervention bias was revealed to be of low risk in all studies, since there was no confounding bias and the selection of participants was explained and well done. At intervention, three studies classified the intervention correctly, but other study did not provide enough information on the various interventions. The post-intervention bias was low to moderate risk, because it lacked information on results and some data was missing.

The quality assessment of the studies included in this systematic review were considered low to moderate risk, reflecting the moderate quality of the systematic review.

## 4. Discussion

Bisphenol A is one of many commercial chemicals found in daily life, with a growth in production of 6% to 10% per year in 2003 [18] and actually (2018) with an annual production of 5 million tonnes in the United States and an annual increase of 13% in Asia [32]. BPA has many applications in manufactured productions such as antioxidants in cosmetics and food, polycarbonate plastics, and dental resin materials. Despite the US Environmental Protection Agency’s (EPA) reference range for acceptable daily BPA exposure being set at <50 µg/kg body weight/day in previous studies, BPA eluted from dental resins has been reported under reference levels [4]. However, temporary Tolerable Daily Intake (TDI) for BPA, calculated by the EPA as well as by the European Food Safety Authority (EFSA), was reduced from 50 μg/kg/day to 4 μg/kg/day in 2015 (EFSA, 2015), increasing the importance of control in release of this compound or even its integration into the composition in various materials.

The time each study was conducted also contributed to the variability of the BPA detected from biological fluid. It is expected higher BPA levels from studies published in 1990 is due to the improvement of resin materials used as dental sealants over time. In this systematic review only studies after 1990 were included, which allows comparison between them.

Olea et al. (1996) initially reported the dissolution of BPA, and the huge problem of its release to human health [29]. The first problem in investigating the side effects of this release through dental materials is that this component is present in everyday life and humans are exposed to numerous sources of BPA. There are, therefore, many confounding factors that can undermine the results of the studies. Most studies take this factor into account and try to minimize it by giving a questionnaire to participants about daily habits and providing some recommendations to decrease BPA exposure from other sources before and during the study. The confounders should be demographic features such as age, gender, socioeconomic conditions, parent’s marital status and country of birth; salivary factors such as salivary flow rate and salivary buffer capacity; and behavioral factors such as frequency of snacking, child’s consumption of sugary drinks (juice, fruit drinks, soda), body mass index (BMI) category, frequency of tooth brushing and other habits with plastic use [6,7,8,33,34].

In this systematic review we analyzed only studies that had samples of saliva, urine and blood as the study object, since the aim was to determine the levels of BPA released and absorbed by the human body. Although, the detection methods between the review studies were similar (gas chromatography and/or high-performance liquid chromatography coupled to mass spectrometry; Enzyme-Linked Immunosorbent Assay; or Flow Cytometry), as was the sample substrate. The BPA- Enzyme-Linked Immunosorbent Assay (ELISA) test is based on the recognition of BPA by specific monoclonal antibodies and the quantitative analysis ranges from 0.05 to 10 μg/L, sensitive enough to detect BPA in field specimens, allowing the use of various types of sample and with the huge advantage of the ease of handling. On the other hand, liquid and gas chromatography coupled with mass spectrophotometry are analytical techniques with high selectivity and sensitivity, although with numerous steps with increased risk of contamination.

As the assays’ measurement units were different with very disparate follow-ups, it was impossible to perform a meta-analysis. However, the quality assessment of the studies, both RCT and cohort studies, demonstrated low risk of bias in most parameters, with some moderate risk of bias, concluding a systematic review with strong clinical evidence.

Some other studies analyze the release of BPA in mouthwash, and concluded that there was an increase of BPA in the first mouthwash after treatment, but that it was lower than the tolerable daily intake limits and decreased in the several follow-ups [35]. However, the authors recommend mouthwash after the application of resins in the oral cavity in order to reduce this immediate increase in BPA [35].

This finding is similar in most studies of the review. The detection of BPA in the saliva was transversal to all the studies that analyzed it, which did not occur in the studies that evaluated urine. Not all of them presented increased values of BPA in the various follow-ups. For those who evaluated saliva, the majority showed an increase in BPA after treatment up to 1 h. From 24 h, a reduction in the amount of BPA present is found, like the control done before the treatment from the first week of follow-up. This is corroborated by another systematic review that addresses only fissure sealants [36]. As in amalgam restorations, where there is no reference to the maximum number of surfaces treated in each patient, there is also no such reference in resin treatments [37,38]. The 1 h increase in BPA levels in saliva was cross-sectional to all studies, even with only one surface treated. However, the number of treated surfaces was a factor that influences this degree of increase [6,10,21,22,23]. Some studies report this was significant from four treated surfaces, which corresponds to about 8 mg for each one when fissure sealants are used [6,10]. Other studies report this considerable increase from six or even 11 [21,22]. Another author referred that this increase is double for each treated surface [23]. From July 2018, dental amalgam should not be used in some population groups, namely children under 15 years of age, and pregnant or breastfeeding women, as regulated by EU. For surfaces treated with resins, there is no specific recommendation, and there exist only a few studies, such as those reported. The scientific community has already started discussion of this problem, but has not yet make clinical recommendations [38].

In addition to these clinical studies, there are several in vitro studies that have similar results, despite the limitations of these types of studies. The ex-vivo study of Malkiewicz et al. in 2015 [39] evaluates the release of BPA from six orthodontic adhesives based on light-cured polymers and detected the presence of BPA in one of them, Resilence, with maximal concentration at 1 h (32.10 μg/mL) [39]. 3M Transbond™ XT (3M, St. Paul, Minneapolis, MN, USA), the most tested orthodontic adhesive, was also evaluated in several studies [40]. It was shown that BPA release of this adhesive increased from day 1 to day 21 with a maximal concentration in 10 mm tip distances with Light Emitting Diode (LED) 20 s curing time and Halogen Light Colour (HLC) 40 s curing time, and after that the levels mostly decreased [40]. The LED group showed less BPA release than HLC group for similar tip distances. To sum up, an increase in tip distance caused greater BPA release and a decrease in degree of monomer to polymer conversion. Sunitha et al. in 2011 [41] had similar results and conclusions [41]. In another in vitro study, Eliades et al. in 2011 [42] found the highest concentration of BPA in Transbond™ XT (3M, St. Paul, Minneapolis, MN, USA) in the 1 month group (2.9 mg/L) [42]. American Dental association (ADA) Science Institute recently analyzed 12 dental sealants and concluded that they showed extremely low BPA release (0.09 nanograms of BPA in four teeth applications) [43]. Becher et al. in 2018 [43] analyzed three pit and fissure sealants: Clinpro^TM^ Sealant (3M, St. Paul, Minneapolis, MN, USA), Delton^®^ (Dentsply Sirona, York, PA, USA) and Helioseal^®^ F (Ivoclar Vivadent Inc., New York, NY, USA). This study compared the leached BPA values obtained by uncured materials and cured materials and it was confirmed that Delton (Dentsply Sirona, York, PA, USA) was the one cured material with BPA levels significantly above the control levels and showed the highest BPA leaching after 24 h (9.6 ± 2.2 ng/mL), in an immersion medium of deionized water without hydroquinone, which is maintained during the following 15 days [43]. These results were similar to clinical studies with this material. However, it is also reported that this fissure sealant resin does not have the ADA seal of acceptance [1,10,27].

As already mentioned, BPA and its derivates have numerous side effects already demonstrated as endocrine disruptive effects, potential estrogenic activity, and as a predisposing factor for several pathologies such as obesity, diabetes and various types of cancer.

BPA is considered a xenoestrogen producing biological outcomes such as the natural hormone, [44,45,46] being considered an endocrine disruptor [47]. Human exposure to endocrine disruptors such as BPA might even begin in fetal development since these compounds traverse the placenta. They may cause interferences in organogenesis, making it more harmful than in adulthood [48]. Since estrogens influence the development and regulation of the female genital tract, perinatal exposure to estrogenic compounds results in morphological and functional alterations of the female genital tract and mammary glands that may predispose the tissue to earlier/higher onset of disease, altered fertility and fecundity, altered lactation, breast cancer, advanced puberty and altered estrous cycles [44,48]. The male genital tract might also be affected and become more predisposed to prostate and testicular cancer as well as a drop in sperm count [44,48]. Recently, reports demonstrated that BPA make hormonal changes, namely on thyroid hormones as well as in testosterone levels in boys [26,49], accelerating maturational changes in girls and increasing the risk of diabetes and breast cancer [26]. Therefore, dental procedures using BPA-containing composite resins or sealants should be practiced with caution in pregnant women. Other in vivo studies with animal models demonstrated some of these adverse effects and confirmed the toxic effect on fertility and reproduction of female mice of monomers such as TEGDMA and BisGMA [50,51]. The BisGMA demonstrated a high embryotoxic/teratogenic effect, due to the molecule structure and/or its higher lipophilic character, allowing passage through the cell membranes and cell organelles and even led to DNA single strand breaks [52].

Human oral cells are among the first to be in contact with eluted substances by means of the use of dental products such as resins or sealants, and are a potential genotoxic risk to humans [53,54,55].

TEGDMA and UDMA genotoxicity is related to the generation of DNA lesions associated with homologous recombination and gene and chromosomal mutation, instead BisGMA and HEMA [53].

More adverse effects might be expected for people working in close contact with these substances and if a causal relationship is confirmed for occupational exposure to xenoestrogens (dentists, dental technicians or workers in the resinous materials manufacturing industry) [44,56,57].

Further studies will be needed relating to the release of BPA and subsequent presence in various body fluids and as modifications in application techniques can influence this presence. Other large-scale studies, such as epidemiological studies, would be important in assessing the presence of BPA in the population, its relationship to dental treatments, and whether it may be a predisposing factor for cardiovascular disease, diabetes, cancer and neurological disease.

## 5. Conclusions

The clinical trials that were included in this systematic review and other in vitro and in vivo in animal model studies found in the literature enable us to really understand about the toxicological potential of BPA and the medical evidence about its pathological effects. The studies report that there is a marked increase in BPA in the first hour after the application of composite resins and fissure sealants. This increase is all the greater as more surfaces are made, with an exponential increase from four surfaces. After 24 hours the levels decrease to the base values.

However, we do not know the quantities at which these damages are effective and if there is a cumulative effect among the various sources or if any of them has an individual potentiating effect. On the other hand, the use of composite resins and fissure sealants is proven to be beneficial for the oral health of patients. Therefore, in response to the PICO question asked, we offer the following recommendations for prudent practice:(1)Resin composition used should be considered since some monomers have more estrogenic effects than others. Bis-GMA is preferred over Bis-DMA. However, the choice becomes difficult because most composite resins and sealants have several different monomers in their composition. The safety data sheets must be as complete as possible so that the percentage of each of the monomers can be evaluated and based on this information for the clinician to make the best choice.(2)Restorations or sealants must be done with a rubber dam to minimize their dissolution in the saliva.(3)To eliminate the last layer of resin unpolymerized by oxygen, a glycerin gel barrier must be placed prior to polymerization or alternatively surface polishing with a pumice or cotton applicator, or at least one air/water spray wash for 30 s should be carried out.(4)The patient should do a 30 s mouthwash after treatment because it is essential to introduce measures to dilute it for better patient safety.(5)Choose photopolymerizable composite resins instead of self-curing ones, for example in the application of an orthodontic appliance.(6)Special attention should be given to treatments in children, adolescents and pregnant women due to the high estrogenic and teratogenic level of BPA. For these patients, all the clinical precautions suggested should be taken simultaneously. For pregnant women, the postponement of treatment should be a clinical consideration, especially in the first trimester of pregnancy.(7)Do the smallest possible number of treatments in a single appointment to reduce the potential increase in BPA release. Do not perform more than four treatments per appointment, both restorations and sealants.

## Figures and Tables

**Figure 1 ijerph-16-01627-f001:**
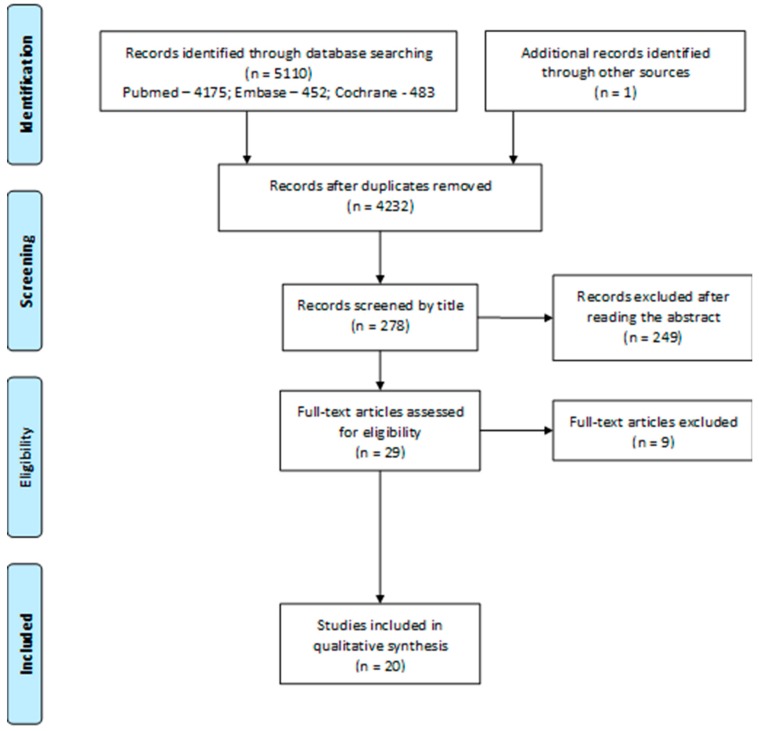
Flow diagram of the study selection.

**Table 1 ijerph-16-01627-t001:** Summary of the included studies on systematic review.

Author/Year	Type of Study	Sample	Type of Treatment	Object of Study	BPA Sources/Materials	BPA Evaluation Methods	Follow- Up	Results	Notes
Kingman A et al., 2013 [17]	RCT	*n* = 172 (264 teeth = 120 mg resin/patient) Mean age— 43.9 ± 1.1 years	Composite restorations	Saliva (*n* = 151) Urine (*n* = 171)	G1—BPA G2—BPAHPE G3—TEGDMA G4—BADGE G5—Bis-DMA G6—Bis-GMA	Liquid Chromatography System Mass Spectrometer (Limits of detection: personnel derived statistically valid lower reporting limits)	T0—0–<1h (*n* = 151) T1—1–8 h (*n* = 44) T2—9–30 h (*n* = 107)	Saliva (ng/mL): T0; T1; T2 G1—0.43; 0.64; 0.41 G2—0.98; 120.50; 1.28 G3—0.70; 4.67; 0.75 G4—1.34; 1.30; 1.43 G5—0.62; 0.66; 0.58 G6—3.09; 198.65; 3.24 Urine (ng/mL): T0; T1; T2 G1—1.75; 1.05; 2.38 G2—1.09; 1.25; 1.10 G3—0.47; 0.50; 0.46 G4—0.61; 0.62; 0.62 G5—0.26; 0.23; 0.31 G6—3.24; 2.68; 3.60	Lack of information on the dental adhesives used. Rubber dam effect on BPA concentrations was studied. Statistically significant (before/after) with BPAHPE and Bis-GMA
Kang, Y et al., 2011 [18]	RCT	*n* = 22 Mean age—21.5 years	Bonding of lingual retainers	Saliva Urine	Adper Single Bond2 + G1—Filtek Flow (Bis-GMA+TEGMA) G2—Z250 Universal Restorative (Bis-GMA + BIS-DEMA)	Liquid Chromatography System (Limit of Detection: 0.5 ng/mL)	T0—Before treatment T1—30 min* T2—1 day T3—1 week T4—1 month	Saliva (ng/mL): (*n* = 20) T0: G1 (0 ± 0.0000); G2 (0.8389 ± 2.2685) (*n* = 20) T1: G1 (2.3211 ± 2.2000); G2 (7.2676 ± 68186) (*n* = 19) T2: G1 (0.5525 ± 1.5627); G2 (0.3684 ± 1.2217) (*n* = 19) T3: G1 (0.0914 ± 0.2743); G2 (0.8502 ± 2.3258) (*n* = 20) T4: G1 (0 ± 0.0000); G2 (0 ± 0.0000) Urine (ng/mL): (*n* = 22) T0: G1 (0.7974 ± 1.6509); G2 (0.3284 ± 0.7064) (*n* = 22) T2: G1 (0.5897 ± 1.1459); G2 (4.1116 ± 6.3120) (*n* = 19) T3: G1 (0.6987 ± 1.0267); G2 (3.3291 ± 6.2734) (*n* = 22) T4: G1 (2.8113 ± 4.0100); G2 (0.7988 ± 1.5626)	*Only saliva was collected after 30 min. Water irrigation/pumice effect on BPA concentration was studied.
Zimmerman-Downs, J. et al., 2010 [19]	RCT	*n* = 30 (18–40 years)	Dental sealants	Saliva Blood	Delton^®^ Pit & Fissure Sealant—Light Cure Opaque 91.2% ADM, 1% EDA G1—Low-dose G2—High-dose	BPA Enzyme Linked ImmunoSorbent Assay (ELISA) (Quantitative analysis ranges from 0.05 to 10 μg/L (ppb))	T0—1 h before treatment After treatment— T1—1–3 h T2—3–24 h	Baseline Salivary BPA (both groups): 0.07–6.00 ng/mL T1—Low dose: 3.98 ng/mL; High dose: 9.08 ng/mL T2—significant decrease to baseline values No statistically significant difference between mean salivary BPA concentration levels in low or high-dose groups at 1 h prior (*p* = 0.4328) or 24 h post (*p* = 0.3283).	Low-dose group: One occlusal sealant application. High-dose group: Four occlusal sealant applications Blood serum did not contain BPA at any point in this study.
Sasaki et al., 2005 [20]	RCT	*n* = 21	Composite restoration	Saliva	G2—Z 100 (Bis-GMA/TEGDMA); Toughwell (Bis-GMA) Beautifil (Bis-GMA/TEGDMA); Xeno CFII (Bis-GMA); Prodigy (Bis-GMA/TEGDMA); Cleafil ST (Bis-GMA/TEGDMA) G1—Progress (UDMA/TEGDMA); Palfique; Matafil Flo (UDMA/TEGDMA); Unifil S(UDMA);	BPA ELISA “EIKEN” Kit (Quantitative analysis ranges from 0.05 to 10 μg/L (ppb))	T0—Before treatment T1—immediately after treatment	G1 T0—0.3–2.0 ng/mL (mean 0.87 ± 0.69 ng/mL); T1—21.0–60.1 ng/mL (mean 32.1 ± 16.27 ng/mL); After gargling—1.6–4.7 ng/mL (3.1 ± 1.47 ng/mL) G2 mean <40 ng/mL or lower	Water effect on BPA concentration was studied.
Chung et al., 2012 [21]	RCT	*n* = 495 (8–9 years)	Resin, sealant and resin composites	Urine	Without information	Classification by the number of composite resins and sealant surfaces (0, 1–5, 6–10 and 11+) Liquid Chromatography System (Creatinine-adjusted urinary BPA) (Limit of Detection: Without information)	Without information	Mean of surfaces—10.07 ± 8.44 The mean creatinine-adjusted urinary BPA concentration was 2.08 ± 3.81 μg/g creatinine; Children with 11 or more composite resin surfaces—2.67 μg creatinine Resin—0 surfaces (−0.65); 6–10 surfaces (−0.43); +11 surfaces (1.02) μg/g Sealants—0 surfaces (0.22); 6–10 surfaces (−0.63); +11 surfaces (9.13) μg/g Resin composites—0 surfaces (0.06); 6–10 surfaces (−0.49); +11 surfaces (2.68) μg/g	
Fung et al., 2000 [10]	RCT	*n* = 40 (20–55 years age)	Dental sealant	Saliva Blood	Delton Opaque Light cure Pit and fissure sealant	High-pressure liquid chromatography (HPLC) (Limit of Detection: 5 ppb)	Baseline—before treatment After treatment— 1 h, 3 h, 1 day, 3 days and 5 days	Low-dose—8 mg dental sealant on 1 surface High-dose—32 mg of sealant (8 mg on each four surfaces) BPA in some saliva specimens (5.8–105.6 ppb) collected at 1 h and 3 h. The BPA was not detectable beyond 3 h or in any of the serum specimens. For the 1- and 3-h saliva samples, the BPA concentration in the high-dose (32 mg) group was significantly greater than in the low-dose (8 mg) group (*p* < 0.05).	In the high-dose group, there was a significant decrease in saliva BPA concentrations from 1 to 3 h (*p* < 0.01).
Maserejian et al., 2016 [22]	RCT	*n* = 91 (Age—3–17 years) 1 s—43.9% 2 s—25.3% 3 s—17.6% 4 s—8.8% 6 s—2.2% 8 s—2.2%	Adhesive Resin composite Dental sealant	Urine	G1—resin restoration with adhesive and composite (69.2%) + fissure sealant if needed (38.5%). Z100 restorative (Bis-GMA) + Optibond bonding agent (Bis-GMA) + Embrace (Bis-GMA)	Solid-phase extraction-high performance liquid chromatography isotope-dilution tandem mass spectrometry (Limit of Detection: 0.1 ng/mL)	Baseline—before treatment—T0 After treatment— 24 h—T1 14 days—T2 6 months—T3	(0 to 3 surfaces) ng/mL T0–T1(*n* = 89)—3.33 ± 3.84 to 5.04 ± 9.94 (+51.4%) T0–T2 (*n* = 81)—3.45 ± 3.97to 2.95 ± 4.09 (−14.5%) (3 to 5 surfaces) ng/mL T0–T1 (*n* = 26)—3.45 ± 3.35 to 2.86 ± 3.62 (−17.1%) T0–T2 (*n* = 15)—3.21 ± 2.36 to 2.93 ± 3.55 (−0.4%) (+6 surfaces) ng/mL T0–T1 (*n* = 3)—1.03 ± 0.53 to 3 ± 2.91 (+191.3%) T0–T2 (*n* = 5)—1.91 ± 1.38 to 1.64 ± 1.28 (−14.1%) T0–T3 (*n* = 77)—3.07 ± 3.01 to 3.36 ± 4.66 (+9.4%)	In 51.6% participants rubber dam was used.
McKinney et al., 2014 [9]	RCT	*n* = 1001 (Age—6–19 years)	Resin-based dental sealants and composites	Urine	Bisphenol A-glycidyl methacrylate	Without information	Without information	Lowest quartile had BPA concentrations of 0.3–1.9 ng/mL. Highest quartile had mean BPA concentrations of 7.3 to 149 ng/mL Children with 7–16 sealants—BPA +25%; +10 sealants— +11% higher (BPA) Children with 7–42 restorations had (BPA) 20% higher	It is not possible to conclude on the increase of urinary concentrations of BPA after the placement of sealants or restorations, nor is the time or sources of other exposures to BPA known
Lee et al., 2017 [23]	RCT	*n* = 30 (Mean age—40)	Composite Resin	saliva	Filtek Z350 XT	Ecologiena^®^ supersensitive BPA ELISA Kit (Quantitative analysis ranges from 0.05 to 10 μg/L (ppb))	Before treatment—T0; After treatment: 5 min—T1 and 7 days—T2	BPA (ug/L) in saliva T0 *n* = 30, Mean—0.15 ± 0.42 (0 teeth—*n* = 20, Mean—0.18 ± 0.51; 1 tooth or more—*n* = 10, Mean—0.09 ± 0.09) T1 *n* = 30 Mean—3.64 ± 2.32 (1 tooth—*n* = 13, Mean—2.67 ± 2.32; 2 teeth or more—*n* = 17, Mean—4.38 ± 2.10) T2 *n* = 30 Mean—0.59 ± 1.27 (1 tooth—*n* = 13, Mean—0.32 ± 0.36; 2 Teeth or more—*n* = 17, Mean—0.79 ± 1.65)	The level of salivary BPA was not significantly influenced by the number of teeth or surfaces of teeth previously treated with the filling of composite resin.
Han et al., 2012 [6]	Case-control study	*n* = 302 *n*—62 with >4 surfaces; 62 controls Ages: Control—10.13 ± 2.14 Ages Experimental group—10.03 ± 2.09	Dental sealant/resin	saliva	Without information	Ecologiena^®^ supersensitive BPA ELISA Kit (Quantitative analysis ranges from 0.05 to 10 μg/L (ppb))	Without information	0 surface (*n* = 62 teeth) BPA, mean ± SD 0.42 ± 0.38 >4 surfaces (*n* = 62 teeth) BPA, mean ± SD 0.90 ± 1.80 Children with four or more surfaces with sealants or composite resin had higher BPA salivary values *p* = 0.239; after adjusting for confounding variables *p* = 0.026. Salivary BPA level was in the range of doses detectable and there may be a relationship between salivary BPA level and dental sealant/resin in Korean children.	Age, gender, salivary flow rate, salivary buffer capacity, snack frequency and brushing frequency were selected as confounding factors.
Moreira et al., 2017 [5]	RCT	*n* = 20 (Age 12–18)	Adhesive and composite resin	Urine Saliva	Transbond XT system (adhesive and resin)	Gas Chromatograph mass spectrometer (Limit of Detection: Without information)	T0—before treatment: T1—30 min. T2—24 h T3—1 week T4—1 month	BPA in saliva (ng.g^−1^): T0—0.56 ± 0.06 T1—1.04 ± 0.28 (*p* < 0.05) T2—0.64 ± 0.21 T3—0.76 ± 0.33 T4—0.61 ± 0.16 BPA in urine (ng.g^−1^): T0—2.17 ± 0.93 T1—5.04 ± 2.47 (*p* < 0.05) T2—4.22 ± 2.07 (*p* < 0.05) T3—3.05 ± 1.61 T4—2.17 ± 0.93	Bonding brackets with the Transbond XT orthodontic adhesive system resulted in an increase in BPA levels in saliva and urine. The levels were significant, but lower than the reference dose for daily intake and decreased with time.
Berge et al., 2017 [24]	RCT	*n* = 40 (Age 20–35) Test G-20 (with six or more restorations) Control G-20	Resin composite	Saliva	G Test—with six or more restorations) G Control—without intervention Score 1—class I and V restorations Score 2—small class II, III and IV restorations Score 3—extensive class II	Triple quadrupole linear ion trap mass spectrometer coupled to a liquid chromatography system (Limit of Detection: 0.1 ng/mL)	Without information	BPA total (ng/mL): G Test—0.11 (*p* = 0.302) G Control—no detection BPA free (ng/mL): G Test—0.12 (*p* = 0.044) G Control—no detection. Total BPA concentration was higher but not significantly higher in G test compared to G control There was no significant correlation between the size and number rest. and free BPA.	The presence of restorative material based on dental polymers was associated with the slightly elevated concentration of free BPA in saliva.
Raghavan et al., 2017 [25]	RCT	*n* = 45	Retainer post-fixed orthodontic	Saliva	G1—Vacuum-formed retainer G2—Hawley retainer fabricated by heat cure G3—Hawley retainer fabricated by chemical cure	HPLC (Limit of Detection: Without information)	Before placement—T0 After placement— T1—1 h T2—7 days T3—30 days	BPA levels greater in G1 and G3 G1—increase BPA from T0 to T1 (+1.20 ppm); increase BPA from T1 to T2 (+1.18 ppm); decrease BPA from T2 to T3 (−2.18 ppm) G2 and G3—increase BPA from T0 to T1; decrease BPA from T1 to T2; increase BPA from T2 to T3	
Joskow et al., 2006 [1]	Prospective Cohort	*n* = 15 (86 teeth nG1 = 30; nG2 = 56)	Dental sealants (7.36 mg/sealant) (40.35 mg/patient)	Saliva Urine	G1—Helioseal F G2—Delton Light Cure Opaque	Gas chromatograph-high resolution mass spectrometer (Limit of Detection: 0.1 ng/mL)	T0—Pretreatment T1—Immediately after treatment T2—1 h after treatment	G1—BPA levels 5.5 μg G2—BPA levels 110 μg Saliva (ng/mL) G1 (T0—0.22 ± 0.03; T1—0.54 ± 0.45; T2—0.21 ± 0.03) G2 (T0—0.34 ± 0.19; T1—42.8 ± 28.9; T2—7.86 ± 12.73) Urine (ng/mL) G1 (T0—2.12 ± 0.93; T1—7.26 ± 13.5; T2—2.06 ± 1.04) G2 (T0—2.6 ± 1.4; T1—27.3 ± 39.1; T2—7.34 ± 3.81)	Delton Light Cure Opaque is a sealant without the ADA seal of Acceptance
Manoj et al., 2018 [26]	RCT	*n* = 4 (Age—13–30 years)	Adhesive and resin composites (brackets bonded)	Saliva	G1—Unite no-mix adhesive G2—Transbond Xt light cure adhesive	HPLC/mass spectrometry method (Limit of Detection: 0.5 ng/mm3)	T0—before treatment T1—30 min. T2—1 day T3—1 week T4—1 month	G1—μg/mL T0—0.0 ± 0.0; T1—19.6 ± 8.0; T2—5.0 ± 1.3; T3—4.0 ± 1.3; T4—1.2 ± 0.8 G2—μg/mL T0—0.0 ± 0.0; T1—11.2 ± 4.2; T2—3.1 ± 1.0; T3—2.0 ± 1.0; T4—0.6 ± 0.32	
Arenholt-Bindslev et al., 1999 [27]	RCT	*n* = 8 (Age—20–23 years)	Fissures sealants	Saliva	G1—Visio-Seal G2—Delton LC pit and fissure sealant Clear	HPLC (Limit of Detection: 0.1 ppm; Quantitation limit: 0.3 ppm) Estrogenic assay (spectrophotometrically)	T0—before treatment T1—1 min. T2—1 h T3—24 h	G1—ppm T0, T1, T2, T3—≤0.1 G2—ppm T0—≤0.1; T1—1.43; T2, T3—≤0.1 Dose-range relevant—maximum effect level 1.56 ppm)	BPA present in saliva after treatment with Delton LC. After 1 h neither BPA nor estrogenic activity could be detected.
Michelsen et al., 2012 [28]	RCT	*n* = 10 (Mean age— 54.5 ± 4.1)	Resin composites + Resin Adhesive (score size 0–3) (use rubber Dam)	Saliva	Filtek Z250 (TEGDMA 1%–5%, Bis-GMA 1%–5%, Bis-EMA 5%–10%, and UDMA 5%–10%) Scotchbond 1 (Bis-GMA 10%–30%, HEMA 5%–25%, and dimethacrylates 7%–28%)	Gas Chromatography combined with mass spectrometer Liquid Chromatography combined with mass spectrometer (Mass range Detection: 5–350 m/z)	T0—before treatment T1—10 min. T2—24 h T3—7 days	T1— HEMA—0.068 μg/mL^−1^ TEGDMA—0.00 μg/mL^−1^ Bis-GMA—2.149 μg/mL^−1^ UDMA—0.188 μg/mL^−1^ T0, T2 and T3—0.00 μg/mL	Patients were also asked not to use lipstick or lip balm, not to chew chewing gum, and not to eat pastilles or candy before sampling.
Olea et al., 1996 [29]	RCT	*n* = 18 (Range age—18–25)	Fissures sealants	Saliva	Bis-GMA (50 mg/Patient)	HPLC Gas Chromatography with Mass Spectrometer (Limit of Detection: Without information)	T0—before treatment T1—1 h	T1—90–931 μg/mL	unpolymerized material collected during 1 h after treatment never exceeded 2% of the total of sealant
Maserejian et al., 2014 [30]	Prospective Cohort	*n* = 534 (Age 6–10)	G1 Sealant G2 Preventive resin G3 Resto. on primary teeth G4 Resto on permanent teeth	Blood	G1 Ultraseal XT (bisGMA, diurethane dimethacrylate) G2 Revolution (bisGMA) G3 Dyract AP compomer (UDMA, trimethacrylate resins) G4 Z100 composite 3M ESPE (St. Paul, MN, USA) bisGMA, TEGDMA) G5 amalgam	Flow Cytometry (immune function) (i) white blood cell enumeration, (ii) T cell responsiveness, (iii) B cell responsiveness, and (iv) neutrophil and monocyte responsiveness.	T0—before treatment T1—5–7 days T2—6 months T3—12 months T4—18 months T5—5 years	Positive association—non-flowable BisGMA-based composites—changes in B cell activation (indicating increased activation), was present at 6 months and 1 year, but not at the 5-year visit. BisGMA-based flowable (sealant or non-flowable composites increased, monocyte and neutrophil functions were decreased at 6 months and 1 year, but not at year 5.	
Trachtenberg et al., 2014 [31]	Prospective Cohort	*n* = 534 (Age 6–10)	G1 Sealant G2 Preventive resin G3 Resto. on primary teeth G4 Resto on permanent teeth	Urine Blood	G1 Ultraseal XT (bisGMA, diurethane dimethacrylate) G2 Revolution (bisGMA) G3 Dyract AP compomer (UDMA, trimethacrylate resins) G4 Z100 composite 3M ESPE (St. Paul, MN, USA) bisGMA, TEGDMA) G5 amalgam	Flow Cytometry (renal function) Gamma-glutamyl transpeptidase (gamma-GT) Albumin and N-acetyl-β-D-glucosaminidase (NAG)	T0—before treatment T1—5–7 days T2—6 months T3—12 months T4—18 months T5—5 years	5 years Composite restorations on permanent teeth was 10.4 ± 17.0 surface-years (range 0–15.1), to compomer restorations on primary teeth was 11.8 ± 18.1 surface-years (range 0–16.7), and to flowable composite sealants and PRRs was 39.9 ± 21.1 surface-years (range 25–54). There was no evidence that composite treatments were associated with impaired renal function.	OR of high albumin excretion decreased with increased exposure to composite restorations on permanent teeth OR of high NAG decreased with increased exposure to dental sealants and PRRs

RCT—randomized controlled trial; G—group; T—follow-up time; s—surface; BPA—Bisphenol A; BPAHPE—Bisphenol A and bis(2,3-hydroxyphenyl)ether; TEGDMA—triethylene glycol dimethacrylate; BADGE—bisphenol A diglycidyl ether; Bis-DMA—bisphenol A-dimethacrylate; Bis-GMA—bisphenol A-glycidyl methacrylate; Bis-DEMA—bisphenol Apolyethylene glycol diether dimethacrylate; ADM—aromatic and aliphatic dimethacrylate monomers; EDA—ethyl-p-dimethyl-aminobenzoate; ELISA—Enzyme Linked ImmunoSorbent Assay; UDMA—urethane dimethacrylate; DGEBA—diglycidyl ether of bisphenol A; Bis-EMA—ethoxylated bisphenol-A dimethacrylate; Bis-MA—2,2-bis[4-(methacryloxy)phenyl]-propane; HPLC—High-pressure liquid chromatography; ELISA—enzyme-Linked Immunosorbent Assay; ppb—parts per billion.

**Table 2 ijerph-16-01627-t002:** Evaluation of quality assessment of randomized controlled trial (RCT) studies of the systematic review.

Random Sequence Generation (Selection Bias)	Allocation Concealment(Selection Bias)	Blinding of Participants and Personnel (Performance Bias)	Blinding of Outcome Assessment (Detection Bias)	Incomplete Outcome Data(Attrition Bias)	Selective Reporting(Reporting Bias)	Other Bias	
	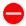						Kingman et al., 2012 [17]
							Kang et al., 2011 [18]
				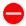			Zimmerman-Downs et al., 2010 [19]
							Sasaki et al., 2005 [20]
		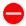					Chung et al., 2012 [21]
		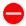					Fung et al., 2000 [10]
		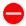					Maserejian et al., 2016 [22]
		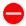					McKinney et al., 2014 [9]
		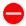					Lee et al., 2017 [23]
		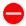					Moreira et al., 2017 [5]
		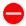					Berge et al., 2017 [24]
		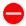					Raghavan et al., 2017 [25]
		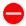					Manoj et al., 2018 [26]
		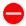					Arenholt-Bindslev et al., 1999 [27]
							Michelsen et al., 2012 [28]
							Olea et al., 1996 [29]


 Low risk of bias; 

 Unclear risk of bias; 
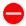
 High risk of bias.

**Table 3 ijerph-16-01627-t003:** Evaluation of quality assessment of non-randomized studies of the systematic review.

Pre-Intervention		At Intervention	Post-Intervention				Non-Randomized Studies of Interventions
Bias Due to Confounding	Bias in Selection of Participants into the Study	Bias in Classification of Interventions	Bias Due to Deviations from Intended interventions	Bias Due to Missing Data	Bias in Measurement of Outcomes	Bias in Selection of the Reported Result	
Y/PY/PN/**N**/NI	Y/PY/**PN**/N/NI	Y/PY/PN/**N**/NI	Y/PY/PN/**N**/NI	Y/**PY**/PN/N/NI	Y/PY/PN/**N**/NI	Y/PY/**PN**/N/NI	Han et al., 2012 [6]
Y/PY/PN/**N**/NI	Y/PY/PN/**N**/NI	Y/**PY**/PN/N/NI	Y/PY/PN/**N**/NI	Y/PY/PN/**N**/NI	Y/**PY**/PN/N/NI	Y/PY/PN/**N**/NI	Joskow et al., 2006 [1]
Y/PY/**PN**/N/NI	Y/PY/PN/**N**/NI	Y/PY/**PN**/N/NI	Y/PY/**PN**/N/NI	Y/PY/PN/**N**/NI	Y/PY/**PN**/N/NI	Y/PY/PN/**N**/NI	Maserejian et al., 2014 [30]
Y/PY/**PN**/N/NI	Y/PY/PN/**N**/NI	Y/PY/**PN**/N/NI	Y/PY/**PN**/N/NI	Y/PY/PN/**N**/NI	Y/PY/**PN**/N/NI	Y/PY/PN/**N**/NI	Trachtenberg et al., 2014 [31]
Low risk	Low risk	Moderate risk	Low risk	Moderate risk	Moderate risk	Low-risk	RISK OF BIAS JUDGEMENTS

Risk of bias—(Y) Yes; (PY) probably yes; (PN) probably no; (N) no; (NI) no information; Choice of bias for each study is bolded.

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
