# Peer review of "Once Resin Composites and Dental Sealants Release Bisphenol-A, How Might This Affect Our Clinical Management?—A Systematic Review"

_ijerph, 2019, doi:10.3390/ijerph16091627_

Round 1
Reviewer 1 Report
This review highlights the importance of minimizing exposure risks associated with using certain resins as dental sealants.
Minor comments:
1. Lines 55-57 should be moved to discussion section
2. No ID122657 was found in PROSPERO
3. The limit of detection of each method used in 20 studies should be provided in the table. The pros and cons of each detection method should be discussed.
4. The time each study was conducted also contributed to the variability of the BPA detected from biological fluid. It is expected higher BPA levels from studies published in 1990 due to the improvement of resin materials used as dental sealants over time. This should be discussed.
5. Lines 183-184: "In the only study evaluating estrogenic activity, an increase immediately after treatment from 183 0.1 ppm to 1.43 ppm was observed, with only one type of fissure sealant (Delton®) decreasing to 184 levels below 0.1 ppm after 24 hours.[27]". This statement needs to be rephrased, because concentrations of BPA per se have little or no relation to the estrogenic activity.
6. Since all in vitro studies were excluded based on the inclusion and exclusion criteria, Appendix A should not be included, and the relevant information in discussion section should be removed. Further, studies like Małkiewicz et al, 2015 should be considered as ex vivo rather than in vitro study.
Author Response
This review highlights the importance of minimizing exposure risks associated with using certain resins as dental sealants.
Minor comments:
1. Lines 55-57 should be moved to discussion section
R: The alterations were made in the manuscript.
2. No ID122657 was found in PROSPERO
R: Sorry, but de ID has an error. The correct number is 122957. We had some incomplete record fields of PROSPERO. We have already completed this information, and the registration is complete. The alteration of the number was made in the manuscript.
3. The limit of detection of each method used in 20 studies should be provided in the table. The pros and cons of each detection method should be discussed.
R: The alterations were made in the manuscript.
4. The time each study was conducted also contributed to the variability of the BPA detected from biological fluid. It is expected higher BPA levels from studies published in 1990 due to the improvement of resin materials used as dental sealants over time. This should be discussed.
R: The alterations were made in the manuscript.
5. Lines 183-184: "In the only study evaluating estrogenic activity, an increase immediately after treatment from 183 0.1 ppm to 1.43 ppm was observed, with only one type of fissure sealant (Delton®) decreasing to 184 levels below 0.1 ppm after 24 hours.[27]". This statement needs to be rephrased, because concentrations of BPA per se have little or no relation to the estrogenic activity.
R: The alterations were made in the manuscript.
6. Since all in vitro studies were excluded based on the inclusion and exclusion criteria, Appendix A should not be included, and the relevant information in discussion section should be removed. Further, studies like Małkiewicz et al, 2015 should be considered as ex vivo rather than in vitro study.
R: We remove the appendix A as suggested by the reviewer, but some of the conclusions of this studies were included in the discussion section. Some studies were changed and considered as ex-vivo, as suggested.

Reviewer 2 Report
The manuscript “Resin composites and dental sealants release bisphenol-A. How might this affect our clinical management? - a systematic review.” summarized the results related to BPA release from preventive and reparative dentistry materials. The analysis methods, exposure levels, possible health risks and recommendations are all included. I think it can be accepted after minor revision.
1. Many studies are focused on the toxicological risks of BPA which have been commonly understood. In the discussion section, the authors described a lot about the biological risks of BPA which is not attractive. I suggest to reduce this section and add some results on transformation/conversion process of BPA in body to attract more attention for this review.
2. Title: the period is not needed.
3. Line 58, add references.
4. Line 67 and other abbreviations: The full names should be given when they first appear in the manuscript.
5. Line 72, add references.
6. The exclusion criteria in line 106 is in contradiction with the description in line 337.
7. Line 132, what is “PROSPERO”?
8. Use BPA after Line 49 instead of bisphenol A and bisphenol-A, and unify bisphenol A and bisphenol-A.
9. Line 175, is the immune function change related to BPA exposure in resin composites and dental sealants?
10. Line 218, is there data in the last three years?
11. Line 237, spectroscopy should be spectrometry.
12. Line 270, of should be from.
13. Lin 275, what is involved in “conversion”?
14. Line 276, highest concentration of “BPA” ?
15. Line 286, what is ADA?
16. Conclusion: this section should include the main topic of this manuscript. The recommendations could be placed in an individual section.
Author Response
REVIEWER 2
The manuscript “Resin composites and dental sealants release bisphenol-A. How might this affect our clinical management? - a systematic review.” summarized the results related to BPA release from preventive and reparative dentistry materials. The analysis methods, exposure levels, possible health risks and recommendations are all included. I think it can be accepted after minor revision.
1. Many studies are focused on the toxicological risks of BPA which have been commonly understood. In the discussion section, the authors described a lot about the biological risks of BPA which is not attractive. I suggest to reduce this section and add some results on transformation/conversion process of BPA in body to attract more attention for this review.
R: Alterations were made in the manuscript.
2. Title: the period is not needed.
R: Alterations were made in the manuscript.
3. Line 58, add references.
R: Alterations were made in the manuscript.
4. Line 67 and other abbreviations: The full names should be given when they first appear in the manuscript.
R: Alterations were made in the manuscript.
5. Line 72, add references.
R: Alterations were made in the manuscript.
6. The exclusion criteria in line 106 is in contradiction with the description in line 337.
R: Alterations were made in the manuscript.
7. Line 132, what is “PROSPERO”?
R: PROSPERO is an International prospective register of systematic reviews, recommended by PRISMA.
8. Use BPA after Line 49 instead of bisphenol A and bisphenol-A, and unify bisphenol A and bisphenol-A.
R: Alterations were made in the manuscript.
9. Line 175, is the immune function change related to BPA exposure in resin composites and dental sealants?
R: No. Alterations were made in the manuscript in to order clarify this question.
10. Line 218, is there data in the last three years?
R: Yes. The alterations were made in the manuscript.
11. Line 237, spectroscopy should be spectrometry.
R: Alterations were made in the manuscript.
12. Line 270, of should be from.
R: Alterations were made in the manuscript.
13. Lin 275, what is involved in “conversion”?
R: Alterations were made in the manuscript.
14. Line 276, highest concentration of “BPA” ?
R: Alterations were made in the manuscript.
15. Line 286, what is ADA?
R: ADA is American Dental Association. This alteration was made in the manuscript.
16. Conclusion: this section should include the main topic of this manuscript. The recommendations could be placed in an individual section.
R: Alterations were made in the conclusion section of the manuscript. It has not created a new section of clinical implications because it is not part of the journal's layout. However, if you understand that the division is more relevant, we can change and create a new section.
